# Role of Sonographic Second Trimester Soft Markers in the Era of Cell-Free DNA Screening Options: A Review

Cristina Taliento [1], Noemi Salmeri [2], Pantaleo Greco [1] and Danila Morano [1,*]

1   Department of Medical Sciences, Institute of Obstetrics and Gynecology, University of Ferrara,
    40120 Ferrara, Italy; cristina.taliento1@gmail.com (C.T.); pantaleo.greco@unife.it (P.G.)
2   Gynecology/Obstetrics Unit, IRCCS San Raffaele Scientific Institute, 20132 Milan, Italy; salmeri.noemi@hsr.it
*   Correspondence: moranodanila@gmail.com; Tel.: +39-0532-239517

**Abstract:** Soft markers are sonographic structural, nonspecific signs with little pathological significance, often transient, usually considered as normal variants. However, they may also be associated with chromosomal abnormalities. The most widely examined soft markers include absent or hypoplastic nasal bone (NB), intracardiac echogenic focus (IEF), ventriculomegaly (VM), thickened nuchal fold (NF), choroid plexus cyst (CPC), echogenic bowel, short long bones, and urinary tract dilation (UTD). Although the use of noninvasive prenatal testing (NIPT) has been spreading quickly in maternal–fetal medicine, it is not a diagnostic test and it still remains unavailable or cost-prohibitive for most of the population in many countries. After normal screening test results in the first trimester, there is no uniform consensus regarding the clinical significance of isolated soft markers for aneuploidy. Nowadays, the search for soft markers in an ultrasound is still part of clinical evaluation, and the interpretation of these findings is often a matter of debate. In the present review, we summarize the recent literature about the role of soft markers in the era of NIPT and propose an overview of the different clinical guidelines.

**Keywords:** soft markers; aneuploidy; sonographic markers; ultrasonography; cfDNA; NIPT

## 1. Introduction

The introduction in clinical practice of NIPT using cell-free DNA (cfDNA) circulating in maternal plasma has significantly changed the scenario of prenatal testing. The International Society for Prenatal Diagnosis (ISPD) considered it appropriate to offer NIPT testing as a first choice screening test for all pregnant women [1]. Compared to the first trimester combined test (FCT), the specificity the sensitivity and the positive predictive value of cell-free DNA testing are remarkably high for the more common trisomies, including trisomy 21, 18, and 13 [2]. If NIPT becomes available as a routine procedure, a deep rethinking on the role of the second trimester "genetic ultrasonogram" in prenatal care is needed [3].

At ultrasound assessment, the major structural anomalies include several anomalies such as atrio-ventricular canal or duodenal atresia, the prevalence of which has been found to be significantly higher in chromosomal abnormal than normal fetuses.

In contrast, minor fetal abnormalities, also called "soft markers", are sonographic structural, nonspecific signs with scarce pathological significance. Indeed, such features are usually seen in normal fetus and are not associated with any handicap unless a coexisting chromosomal abnormality is observed [4]. Historically, their detection at ultrasound was first introduced as a screening strategy when the current screening tests were not available. Nowadays, although the interpretation of these findings is often a matter of debate, the search for soft markers during an ultrasound is still part of clinical evaluation.

The most widely examined soft markers include the following: absent or hypoplastic nasal bone (NB), intracardiac echogenic focus (IEF), fetal ventriculomegaly (VM), choroid plexus cyst (CPC), thickened nuchal fold (NF), echogenic bowel, urinary tract dilation

(UTD), and shortened long bones. Current major international guidelines recommend that when one of those sonographic findings is detected during a routine ultrasound examination, an accurate evaluation should be made for the other characteristics of the chromosomal abnormalities known to be associated with that marker. Thus, only in cases where additional abnormalities of a certain syndromic pattern are detected, the risk of a certain chromosomal defect should be considered as substantially increased.

In the case of apparently isolated abnormalities, major international guidelines suggest that further investigation should be based on the type of abnormality detected. In a recent study, Hu et al. suggested that the recognition of sonographic markers on a second trimester ultrasound should serve as a reminder to consider NIPT as a screening option, although the detection of certain minor structural anomalies requires closer pregnancy follow-up [5].

However, NIPT should not count as a diagnostic test since the presence of a chromosomal defect may be limited to the placental tissue. Additionally, false negative and false positive test results may occur because of several factors such as the presence of maternal chromosomal anomalies and/or maternal cancers, low fetal DNA fraction in maternal blood, vanishing twin, confined placental mosaicism (CPM), uniparental disomy, and fetal mosaicisms. However, given the presence of different technologies for fetal DNA evaluation including counting and SNP methods, some specific clinical situations could yield discordant results among various methods. Furthermore, in those settings where no advanced panels for NIPT including the genome-wide approach or yet the search for microdeletions and de novo mutations are available, a chromosomal abnormality other than trisomy 13, 18, and 21 will not be revealed by the most common NIPT. Accordingly, when a fetal structural abnormality is found at ultrasound assessment, diagnostic testing with chromosomal microarray analysis (CMA) should be offered [6,7].

Although the use of NIPT has been spreading quickly in maternal–fetal medicine, it still remains unavailable or cost-prohibitive for most of the population in many countries. Moreover, it can be extremely useful for pregnant women who receive prenatal care beginning in the second trimester, where rapid information about risk may be required [8–11].

Nowadays, the Society for Maternal-Fetal Medicine (SMFM) does not recommend diagnostic prenatal testing for aneuploidy only of the basis of an isolated ultrasound soft marker following a negative NIPT result [12].

In this review, we analyze current literature on this topic aiming to debate if the time of sonographic markers as a screening instrument for chromosomal abnormalities is coming to an end or if there is still space for it in the context of current maternal serum screening and noninvasive prenatal testing (NIPT). We also propose an overview of the different clinical guidelines (SMFM, The American College of Obstetricians and Gynecologists, Perinatal Service BC) for the management of specific soft markers.

## 2. Soft Markers

The management of the most common soft markers at ultrasound assessment following major international guidelines is summarized in Table 1.

### 2.1. Absent or Hypoplastic Nasal Bone

Agenesis or hypoplasia of the NB is one of the most studied sonographic markers for chromosomal abnormalities. It is caused by absent or incomplete calcification of the bone, which is associated with 60% of fetuses with trisomy 21 along with other common aneuploidies such as trisomy 13 (Patau syndrome), trisomy 18 (Edwards syndrome) and Turner syndrome [13–18]. However, the prevalence of nasal hypoplasia ranges from 1% to 10% in chromosomally normal fetuses, according to the ethnic origin of the mothers (Caucasians to African-Caribbeans, respectively). As previously described by Sonek et al., NB evaluated in the midsagittal plane of the fetal profile should be defined as hypoplastic when it is either absent or with a length of less than 2.5 mm [19] (Table 2).

**Table 1.** Management of isolated second trimester sonographic soft markers.

| Soft Marker | Screening Tests | Society for Maternal-Fetal Medicine (SMFM), 2021 | American College Obstetrics and Gynecology (ACOG), 2020 | Perinatal Services BC, 2020 |
|---|---|---|---|---|
| Absent or hypoplastic nasal bone | cfDNA | If negative: no further testing | If negative: no further testing | Genetic counseling. |
| | Serum screen | If negative: no further testing vs noninvasive vs invasive testing for aneuploidy. | If negative: no further testing | Genetic counseling. |
| | No previous screening/other recommendations | Counseling for noninvasive vs invasive testing for aneuploidy | Detailed ultrasound to assess fetal anatomy. Genetic counseling. Offer aneuploidy testing. | Genetic counseling. |
| Echogenic intracardiac foci | cfDNA | If negative: no further testing | If negative: no further testing | If negative: no further testing |
| | Serum screen | If negative: no further testing | If negative: no further testing | If negative ((SIPS/IPS/Quad): no further testing |
| | No previous screening/other recommendations | Counseling for noninvasive tests for aneuploidy | If isolated, aneuploidy tests should be offered. | Offer Quad screening. |
| Mild to moderate ventriculomegaly | cfDNA | Consider cf-DNA for patients who decline diagnostic testing after counseling about the limitations of this approach. If ventriculomegaly is detected, SMFM recommends that diagnostic testing (amniocentesis) with chromosomal micro-array. Independently of the result, offer diagnostic testing | If negative: no further testing | Fetal Diagnosis Service |
| | Serum screen | | If negative: no further testing | Fetal Diagnosis Service |
| | No previous screening/other recommendations | Offer diagnostic test | Detailed anatomic survey. Genetic counseling. Offer diagnostic tests for genetic conditions, CMV, fetal MRI and US in third trimester. | Fetal Diagnosis Service |
| Thickened nuchal fold | cfDNA | If negative: no further testing | If negative: no further testing | |
| | Serum screen | If negative: no further tests vs noninvasive vs invasive tests for aneuploidy | If negative: no further testing | |
| | No previous screening/other recommendations | Counseling for noninvasive vs invasive testing for aneuploidy | Detailed anatomic survey. Genetic counseling. Aneuploidy testing should be offered if not previously performed. | Genetic counseling. |
| Choroid plexus cyst | cfDNA | If negative: no further testing | If negative: no further testing | If negative: no further testing |
| | Serum screen | If negative: no further testing | If negative: no further testing | If negative ((SIPS/IPS/Quad): no further testing Offer Quad screening. |
| | No previous screening/other recommendations | Counseling for noninvasive testing for aneuploidy | Offer aneuploidy testing. | Genetic counseling is recommended only if CPC is seen in combination with other structural abnormalities or FGR. |

**Table 1.** *Cont.*

| Soft Marker | Screening Tests | Society for Maternal-Fetal Medicine (SMFM), 2021 | American College Obstetrics and Gynecology (ACOG), 2020 | Perinatal Services BC, 2020 |
|---|---|---|---|---|
| Echogenic bowel | cfDNA | If negative: no further testing | If negative: no further testing | Genetic counseling. |
| | Serum screen | If negative: no further testing | If negative: no further testing | Genetic counseling. |
| | No previous screening/other recommendations | Counselingfor noninvasive testing for aneuploidy. Third trimester scan for evaluation of fetal growth. | Detailed anatomic evaluation. Genetic counseling. Offer CMV, CF, and aneuploidy testing. Consider follow up US for fetal growth because of the association with FGR. | Genetic counseling. |
| Urinary tract dilation | cfDNA | If negative: no further testing | If negative: no further testing | If negative: no further testing. |
| | Serum screen | If negative: no further testing | If negative: no further testing | If negative (SIPS/IPS/Quad or NIPT): no further testing |
| | No previous screening/other recommendations | No previous screening: counseling for noninvasive testing for aneuploidy. Third trimester ultrasound examination to determine if postnatal pediatric nephrology or urology follow-up is needed. | Offer aneuploidy testing. Repeat US in third trimester to assess need for postnatal imaging. | Offer Quad screening and postnatal renal scan between 5 and 30 days of age. |
| Shortened long bones | cfDNA | If negative: no further testing | If negative: no further testing | If negative: no further testing |
| | Serum screen | If negative: no further testing | If negative: no further testing | If negative (SIPS/IPS/Quad or NIPT): no further testing |
| | No previous screening/other recommendations | Counseling for noninvasive testing for aneuploidy. Third trimester ultrasound examination for reassessment and evaluation of growth. | Offer aneuploidy testing. Consider repeat US in third trimester for fetal growth. | Offer Quad screening. |

The results of a large meta-analysis including 21 studies showed low sensitivity for hypoplasia and agenesis of the nasal bone. yet yielding high specificity. Although the pooled risk estimates are limited by a lack of standardization, small sample size, and high heterogeneity of included studies, similar results were found between the different variants used to define hypoplasia of the nasal bone [20].

Rare genetic disorders such as Wolf–Hirschhorn syndrome (4p-) and Cri du chat syndrome (5p-) have also been associated with this soft marker [21,22]. These conditions can be also detected with microarray testing performed in addition to the fetal karyotype. However, for pregnant people with negative NIPT and hypoplastic or absent NB, SMFM recommends counseling to estimate the risk of Down syndrome and the discussion of options for no further prenatal testing [12].

## 2.2. Intracardiac Echogenic Focus

Echogenic intracardiac focus (EIF) is described as an echogenic spot appearing in single-sided or, rarely, bilateral cardiac cavity (Figure 1). Hyperechogenicity of fetal soft tissue is usually defined as greater to or equal in brightness than the surrounding bone [23]. This marker was observed with an overall frequency of 5.6% of fetuses and 15–30% of fetuses with Down syndrome [24].

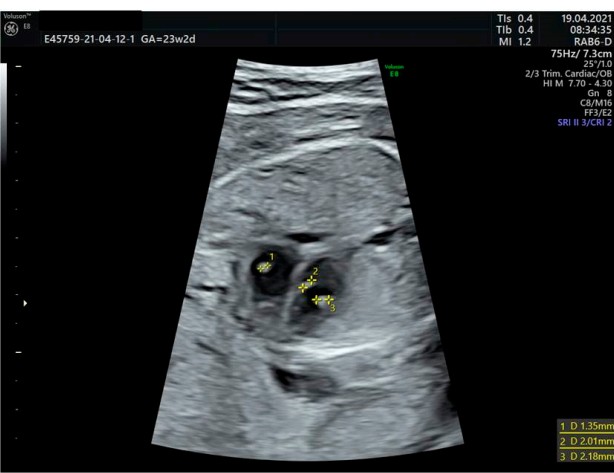

**Figure 1.** Multiple intracardiac hyperechogenic foci.

The results of a meta-analysis including 11 eleven studies showed that EIF increased the risk of trisomy 21 by five- to sevenfold [4]. Other studies have confirmed that fetal EIF is associated with an increased risk of trisomy 21 and further found a significant association also with trisomy 13 (T13). Additionally, it has been demonstrated that fetuses with an intracardiac echogenic focus may have other cardiac defects [25–29].

On the contrary, Bradley et al. found that an isolated echogenic finding appeared to be a normal variant with no associated increased risk for fetal chromosomal abnormalities [30].

These discordant results about associations of EIF may be due to different study designs and populations, different ultrasound techniques, and subjective criteria for diagnosing EIF. However, major international guidelines recommend in case of identification of an EIF further aneuploidy testing by means of noninvasive aneuploidy screening. In cases with a negative result, it is recommended to describe the finding of EIF during the ultrasound scan as not clinically significant or as a benign variant [10–12].

*2.3. Ventriculomegaly*

Ventriculomegaly is a fetal brain anomaly with an overall prevalence estimated to be equal to about 1 over 1000 births. According to the International Society of Ultrasound in Obstetrics (ISUOG) and Gynecology Guidelines, the size of the lateral ventricle should be measured in an axial transventricular plane at the atrium of the posterior horn with calipers placed over the inner edges [31]. The normal width of the fetal lateral ventricular atrium is of a constant size, <10 mm. Ventriculomegaly is categorized as mild (10–11.9 mm), moderate (12–14.9 mm), or severe (≥15 mm) [32,33]. Mild ventriculomegaly is considered a soft marker of abnormal karyotype. The overall frequency of chromosomal defects in fetal ventriculomegaly ranges from 0% to 14.2% with trisomies 13, 18, and 21 and triploidy being the most common associated chromosomal defects [34]. According to the results of the meta-analysis carried out by Agathokleous et al., if ventriculomegaly is present, there is a 3–4-fold increased risk of modifying the pre-test odds for Down syndrome [4]. Moreover, ventriculomegaly can also be caused by structural abnormalities of the central nervous system such as hemorrhage, TORCH infections (toxoplasma and CMV infection), or hypoxic injury [35–38]. Interestingly, this soft marker has been also associated with pathogenic copy number variations (CNVs) identified by CMA [39]. Finally, if identified in the absence of any other soft marker or fetal structural abnormality, mild fetal ventriculomegaly may represent a normal variant [33].

According to this evidence, ACOG and Perinatal Service BC recommend genetic counseling when ventriculomegaly is detected [10–12]. Because of the association between mild ventriculomegaly and fetal aneuploidy or CNVs, SMFM recommends diagnostic testing such as amniocentesis [12].

### 2.4. Thickened Nuchal Fold

The evaluation of nuchal fold thickness during the second trimester ultrasound is one of the most specific and sensitive sonographic soft markers for the identification of fetuses with Down syndrome [40,41]. An NF measurement greater than 6 mm from the outer edge of the occipital bone to the outer skin in the midline has been associated with an increased risk of trisomy 21 [4,42–45].

In a meta-analysis including 56 articles describing 1930 fetuses with trisomy 21 and euploid fetuses, Smith-Bindman et al. showed that an isolated thickened NF was observed in 4% of cases of Down syndrome, whereas this percentage significantly increased to 26% when the thickened NF was observed in addition to other abnormalities. According to these results, the finding of NF thickening in the second trimester increases the probability of trisomy 21 by approximately 17-fold (positive LR, 17; 95% CI, 8–38) [46].

Interestingly, Jelliffe-Pawlowski et al. showed that an increased NF in fetuses with normal karyotype may be associated with increased risk of congenital heart defects (CHDs) [47].

According to the SMFM, for pregnant women with negative cfDNA screening results and isolated thickened NF, no further prenatal screening or testing is recommended [12].

### 2.5. Choroid Plexus Cyst

A choroid plexus cyst (CPC) is a discrete, small cyst of ≥5 mm in one or both choroid plexus(es) (Figure 2). CPC is a common, often transient, sonographic finding, since it can be detected in approximately 0.5–2% of all second trimester pregnancies during an ultrasound examination [48,49].

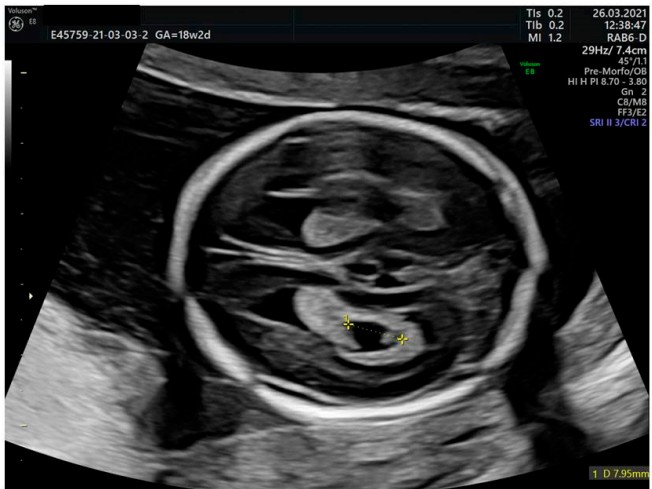

**Figure 2.** Choroid plexus cyst.

Previous studies indicated that the incidence of Edwards syndrome for fetuses with an isolated CPC at ultrasound assessment ranges between 30% and 50% [50–57]. According to the results of a meta-analysis including 13 prospective studies, the risk for Edwards syndrome was about 13 times greater in fetuses with isolated CPC diagnosed in the second trimester. Contrariwise, the probability for Down syndrome did not significantly change from the a priori risk in the presence of an isolated CPC [58]. Remarkably, Goetzinger et al. demonstrated that the positive likelihood ratio for CPC approaches 1, thus confirming that the finding of an isolated CPC is not significantly associated with chromosomal defects [59,60]. However, similarly to other soft markers, when combined with other ultrasound abnormal findings, CPCs do increase the incidence of aneuploidy, with a positive LR as high as 20 for Edwards syndrome [59].

In conclusion, although there is an association between CPCs and aneuploidy, particularly Edwards syndrome, in many cases, they are transient and have a scarce pathological

significance. Accordingly, for patients with negative serum or cell-free DNA screening results and isolated CPC, SMFM does not recommend any further aneuploidy testing or postnatal follow-up [12].

### 2.6. Echogenic Bowel

Echogenic bowel (EB) is an ultrasound finding where the fetal bowel has an echogenicity equal to or greater than that of the adjacent bone. This finding can be seen in 0.2–1.4% of all pregnancies [61,62].

EB is usually a transient soft marker, vanishing when serial scans are performed in the next few weeks [63]. However, persistent EB in the third trimester is more likely to reflect an underlying pathological condition such as aneuploidy, congenital viral infection (CMV infection), cystic fibrosis, severe uteroplacental insufficiency, fetal growth restriction (FGR), or primary gastrointestinal pathology [64]. The association of EB with chromosomal abnormalities, particularly Down syndrome, has been described in several studies. The cause of EB in aneuploidy it is thought to be due to hypoperistalsis with increased water absorption from the meconium. In a retrospective study, the results showed that in 7% of fetuses with trisomy 21, EB was present [65]. However, hyperechogenic bowel is not specific enough nor sensitive as a marker of Down syndrome at second trimester scans [66]. In a recent meta-analysis including 18 studies, D'Amico et al. found that chromosomal anomalies occurred in 3.3% of the fetuses with isolated EB. The most common chromosomal aneuploidy detected in fetuses with isolated EB was Down syndrome (PP: 2.4%, 95% CI 1.2–4.0; 39/1530) and aneuploidies involving the sex chromosomes (PP: 0.7, 95% CI 0.3–1.2; 6/1237), while the incidence of trisomy 13 and 18 was 0.4% and 0%, respectively. Interestingly, cystic fibrosis occurred in 2.2% of the fetuses with isolated echogenic bowel [67].

According to this evidence, for patients with negative serum or cell-free DNA screening results and isolated fetal echogenic bowel, SMFM recommends no further prenatal screening or diagnostic testing [12]. However, tests for the assessment of cystic fibrosis and fetal CMV infection and a third trimester scan for the evaluation of fetal growth are highly recommended.

### 2.7. Urinary Tract Dilation

Urinary tract dilation (UTD) is sonographically described in 1–2% of fetuses [68]. In 2014, a multidisciplinary consensus statement proposed a grading classification system for antenatal UTD based on anterior–posterior renal pelvis diameter, with <4 mm being normal in the second trimester of gestation and <7 mm being normal in the third trimester. In approximately 80% of cases, UTD between 4 and 7 mm in the second trimester of pregnancy is transient, thus vanishing when serial scans are performed, similarly to EB [68]. Some studies have shown that fetal UTD in the presence of other structural abnormalities or other sonographic soft markers is associated with an increased probability of aneuploidy, particularly Down syndrome [69,70]. Interestingly, the prevalence of chromosomal abnormalities in females fetuses is 2 times greater that in males [71]. In order to evaluate the performance of isolated UTD as a soft marker for Down syndrome, Orzechowski and Berghella performed a meta-analysis of 10 studies, showing that UTD increased the risk of Down syndrome from maternal serum screening tests with a pooled positive LR of 2.78 [70].

According to ACOG, if fetal pyelectasis is an isolated finding, aneuploidy testing should be offered if not previously performed. Moreover, for patients with a negative serum or cell-free DNA screening results and an isolated UTD, SMFM recommends no further screening or diagnostic prenatal testing. Additionally, a third trimester ultrasound evaluation to determine the need of postnatal pediatric follow-up is also recommended [10–12].

*2.8. Shortened Long Bones*

Shortened femur is defined as bone length below the 2.5th percentile for gestational age [10]. The detection of shortened long bones has been described as a variant with scarce pathological significance in constitutionally small fetuses [72,73], yet it may also be associated with chromosomal abnormalities, especially trisomy 21 and skeletal dysplasia [73–76]. Interestingly, trisomies 18 and 21 and Turner syndrome are associated with shortened femur and humerus. In a prospective study, Nyberg et al. found that fetuses with trisomy 21 were 5.4 times more likely to exhibit a short humerus than euploid fetuses [74]. Similar results by Benacerraf et al. found that a short humerus carried a risk ratio of 7.9. Rodis et al. reported a risk ratio of 12.8 [75,77].

Therefore, according to ACOG guidelines, in the presence of shortened long bones, screening tests for aneuploidy should be considered if not previously performed [10]. If cfDNA or serum screen are negative, SMFM recommends no further aneuploidy testing. Because of the increased risk of adverse fetal outcomes, such as small gestational age (SGA) fetuses and low birth weight (LBW) in fetuses with shortened humerus or femur or both, the SMFM also highly recommends a third trimester ultrasound for the evaluation of the fetal growth [10,77–80].

**Table 2.** Imaging criteria of ultrasonographic soft markers.

| Soft Markers | Imaging Criteria |
|---|---|
| Absent or hypoplastic nasal bone (NB) | The midsagittal plane of the fetal profile should be defined as hypoplastic when it is either absent or with a length of less than 2.5 mm [20]. |
| Echogenic intracardiac foci (EIF) | Hyperechogenicity of fetal soft tissue greater than or equal in brightness to the surrounding bone [23] |
| Mild to moderate ventriculomegaly (VM) | Measured in an axial transventricular plane at the atrium of the posterior horn with calipers placed over the inner edges [31]. Ventriculomegaly is categorized as mild between 10 and 11.9 mm), moderate between 12 and 14.9 mm [31]. |
| Thickened nuchal fold (NF) | Greater than or equal to 6 mm from the outer edge of the occipital bone to the outer skin in the midline at 15–20 weeks [40,41]. |
| Choroid plexus cyst (CPC) | Cyst of $\geq$ 5 mm in one or both choroid plexus(es) [48,49]. |
| Echogenic bowel (EB) | Fetal intestines display echogenicity equal to or greater than that of the adjacent bone [61,62]. |
| Urinary tract dilation (UTD) | Renal pelvis measuring >4 mm in anterior–posterior renal pelvis diameter up to 20 weeks of gestation [68]. |
| Shortened long bones | Measurement, 2.5th percentile for gestational age [72,73]. |

**3. Discussion**

The soft markers were originally introduced in clinical practice to improve the detection rate of major aneuploidies, especially Down syndrome. Soft markers are common and they are not usually associated with any post-natal impairment. However, as the underlying chromosomal defect may be concealed in those findings, in the 1980s and 1990s, attempts at risk quantification were proposed. Historically, there were two main strategies to try to give a more proper risk assessment of Down syndrome. The first used a simple index scoring system having $\geq$2 as positivity criterion, and a score of 1 is assigned for the soft marker (excluding nuchal fold $\geq$6 mm, which scores 2) [3]. The second was a Bayesian method, named age-adjusted US risk assessment or AAURA, which considers the a priori maternal age-specific risk combined with a quantitative likelihood ratio. In the presence of a previous risk of 1:200 or greater, the test was considered positive.

In the cfDNA era, the role of isolated soft markers has been downsized. All of them do not require any further genetic investigation in the presence of a negative NIPT. However, for some of them, including echogenic bowel, shortened long bones, and urinary tract dilation, NIPT may be not sufficient and a specific antenatal and/or postnatal management

could be required. Again, if NIPT is not available, appropriate genetic counseling is highly suggested, especially when soft markers are detected during the second trimester ultrasound assessment. In fact, according to major international guidelines, routine karyotyping of all pregnancies with these markers would have major implications, both in terms of miscarriage and in economic costs.

In the event of multiple soft markers, ACOG recommends a detailed fetal anatomic ultrasound examination, additional screening, diagnostic testing, and genetic counseling. If not previously performed, patients should be informed regarding the odds of chromosomal abnormalities, and NIPT tests, serum screen testing, or invasive procedures for prenatal diagnosis should be considered. For cases with negative screening test results, ACOG does not recommend any further tests. Otherwise, SMFM recommends offering amniocentesis with CMA when mild ventriculomegaly is detected and offering NIPT tests for only patients who decline invasive diagnostic tests. Similarly, Perinatal Service BC suggests referring the patient to Medical Genetics if a particular marker that increases the risk of trisomy 21 is identified (such as absent nasal bone, increase nuchal thickness, or echogenic bowel). Moreover, consistent with SMFM recommendations, in case of mild ventriculomegaly, Perinatal Service BC suggests referring the patient to the Fetal Diagnosis Center for invasive procedures. According also to SOGC-CCMG Guidelines, in women with a low risk of aneuploidy following first trimester aneuploidy screening, the presence of specific ultrasound "soft markers" associated with chromosomal abnormalities identified during the second trimester ultrasound are not clinically relevant due to poor predictive value and do not warrant further testing.

### 4. Conclusions

Minor fetal abnormalities, also defined as soft markers, are common and they are not usually associated with any post-natal impairment. However, an underlying chromosomal defect may be concealed in those findings, especially when more than one of these structural minor anomalies coexist. According to major international guidelines, routine karyotyping of all pregnancies with these markers would have major implications in terms of miscarriage and economic costs. Thus, appropriate counseling is highly suggested in this setting, and especially when soft markers are detected at second trimester ultrasound assessment, the use of NIPT may provide additional evidence to improve upon counseling and refine the management of pregnancies with such findings. In conclusion, independent of screening tests, sonographic soft markers continue to play a role in the detection of chromosomal abnormalities and should be considered in the complete clinical context in order to supplement the diagnosis of genetic disorders.

**Author Contributions:** Conceptualization, D.M. and C.T.; methodology, D.M.; software, C.T.; validation, N.S., D.M. and P.G. formal analysis, C.T.; investigation, C.T.; resources, C.T.; data curation, C.T. and D.M.; writing—original draft preparation, C.T.; writing—review and editing, D.M., N.S.; visualization, P.G.; supervision, D.M., P.G.; project administration, D.M.; All authors have read and agreed to the published version of the manuscript.

**Funding:** This research received no external funding.

**Institutional Review Board Statement:** Not applicable. No patients were involved in this research.

**Informed Consent Statement:** Not applicable.

**Data Availability Statement:** Not applicable.

**Conflicts of Interest:** The authors declare no conflict of interest.

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
