# Peer review of "Role of Sonographic Second Trimester Soft Markers in the Era of Cell-Free DNA Screening Options: A Review"

_2673-3897, doi:10.3390/reprodmed3030017_

Round 1

Reviewer 1 Report

1) " missing before Soft markers at the start of line 11.

2) Countries on line 18 and 76 should start with small alphabet not capital. 

3) Please add full form for NIPT in abstract where it comes first rather than in introduction. 

4) Remove "if" from line 45

5) Remove extra , from line 49

6) In line 270 in the is repeated twice

7) Apart from few minor corrections in the language, the review is well written and covers the area of soft markers for screening minor physical abnormalities in the fetus in good details.  

Author Response

Reviewer 1

  1. " missing before soft markers at the start of line 11.

Thank you for pointing this out. The error has been corrected on the exact location where the change can be found in the revised manuscript. 

  1. Countries on line 18 and 76 should start with small alphabet not capital.

We thank the reviewer for pointing out this error, which has now been corrected.

  1. Please add full form for NIPT in abstract where it comes first rather than in introduction. 

We thank the reviewer for the suggestion, which we have followed.

  1. Remove "if" from line 45

Many thanks for identifying this. This has been amended.

  1. Remove extra, from line 49

Thank you. We removed it.

  1. In line 270 in the is repeated twice

We thank the reviewer for notice this repetition which has now been deleted.

  1. Apart from few minor corrections in the language, the review is well written and covers the area of soft markers for screening minor physical abnormalities in the fetus in good details. 

Thank you.

Reviewer 2 Report

The Resultats are in a good overview, Bit Not realy new

Author Response

Reviewer 2

  1. The Resultats are in a good overview, Bit Not realy new

Thank you.

Reviewer 3 Report

Dear authors what a nice idea for a paper. Congratulations ive really appreciated it and i think it is very useful. I would just suggest to improve the discussion giving a bit of space to the importance of maternal awareness regarding the prenatal test they undergo, still a huge percentage of women opt for nipt without  beeing aware of the meaning of the test. Therefore I

recommend to read and cite  PMID: 33111167.

otherwise I would recommend it for publication 

Author Response

Reviewer 3

  1. Dear authors what a nice idea for a paper. Congratulations I’ve really appreciated it and i think it is very useful. I would just suggest improving the discussion giving a bit of space to the importance of maternal awareness regarding the prenatal test they undergo, still a huge percentage of women opt for NIPT without being aware of the meaning of the test. Therefore, I recommend to read and cite PMID: 33111167. Otherwise, I would recommend it for publication.

Thank you very much for the kind words and the suggestion. We agree with the reviewer’s comments that the discussion could be enriched with more details about the importance of a proper counseling regarding prenatal test. However, we have chosen to focus on the role of soft markers in the era of cfDNA screening option trying to write a concise review. Thus, we decided to avoid examining in depth those important aspect that require undoubtedly a specific approach.

Reviewer 4 Report

The authors discuss the situation around soft markers for aneuploidy in the second trimester scan in the era of NIPT.

Though many perinatal workers would be interested in this theme, the manuscript seems lacking balanced information or sufficient discussion for publication in Reprod. Med. as a review article.

*Major comments

- Guidelines included is not enough. Others such as ISUOG and SOGC? It is not appropriate to list Perinatal Services BC as equal to SMFM or AJOG. Additionally, SMFM and AJOG is now working together. Thus, the manuscript seems potentially influenced by a bias.

- Partly overlap the contents by Kim et al. (Journal of Genetic Medicine 2018;15:1-7) Thorough literature search performed?

- Many typos and grammatical error.

- Introduction: not easily understandable on the aim of this study.

*Minor comments

- L40 scarce: seems overstatement which could mislead readers to stop screen soft markers.

- L50 guidelines: require references.

- Table 1: difficult to understand what “if negative: none” means.

- L95 the most studied: really? More studied than NT? Need a reference.

- Table 2: Need references.

- L109 Hirshhorn: typo

- L116 “echogenic intracardiac focus” seems more frequently used than “intracardiac echogenic focus”. Keep consistency throughout the manuscript.

- L165 description of NT measurement seems confusing.

- L195 despite: what?

- L277 posterior risk: a technical term in Bayesian theory would not be reader-friendly; require a more easily expression to readers.

Author Response

Reviewer 4

R: The authors discuss the situation around soft markers for aneuploidy in the second trimester scan in the era of NIPT.Though many perinatal workers would be interested in this theme, the manuscript seems lacking balanced information or sufficient discussion for publication in Reprod. Med. as a review article.

Thank you for the comment. More revisions have been made to the relevant parts of the discussion by all the Authors and this section has been expanded after the first submission to this journal. We decided to write a concise review where the main objective was to describe all the ultrasound soft markers in terms of definition, prevalence and association with chromosomal abnormalities. In the original design of the present review, our aim was also to give clear information about the current management suggested by the major international societies.

R: *Major comments

- Guidelines included is not enough. Others such as ISUOG and SOGC? It is not appropriate to list Perinatal Services BC as equal to SMFM or AJOG. Additionally, SMFM and AJOG is now working together. Thus, the manuscript seems potentially influenced by a bias.

We agree and have inserted additional text at the end of the Discussion; “According also to SOGC-CCMG Guidelines in women with a low risk of aneuploidy following first trimester aneuploidy screening, the presence of specific ultrasound “soft markers” associated with chromosomal abnormalities identified during the second trimester ultrasound are not clinically relevant due to poor predictive value and do not warrant further testing”.

We are aware that SMFM and ACOG now working together, but they have produced two different documents with two different Working Group and we found interesting to analyze in details these two guidelines, even if they were quite similar.

- Partly overlap the contents by Kim et al. (Journal of Genetic Medicine 2018;15:1-7) Thorough literature search performed?

Thank you. We performed literature search and we read the review by Kim et al. with interest. However, in our opinion our review focused on management and compared the different guidelines with a specific table to give a quick visual notion. This approach was different from the review performed by Kim et al. We hope that our original work can help many perinatal workers.

- Many typos and grammatical error. 

We accept the reviewer’s comment. Now the typos and grammatical error has been amended as suggested by the reviewers’ comments.

- Introduction: not easily understandable on the aim of this study.

Thank you. We simplified the sentence: “The current review aims to analyze current literature on the role of sonographic markers as a screening instrument for chromosomal abnormalities in the context of current maternal serum screening and noninvasive prenatal testing (NIPT).This paper also proposes an overview of the different Clinical Guidelines (SMFM,  The American College of Obstetricians and Gynecologists, Perinatal Service BC) for management of specific soft markers”.

*Minor comments

- L40 scarce: seems overstatement which could mislead readers to stop screen soft markers.

We accept the reviewer’s comment. We certainly did not mean to imply that the scarce pathological significance means someone will inevitably stop to screen soft markers. However, we change the sentence with “little pathological significance”.

- L50 guidelines: require references.

We agree and have now amended the text with the references.

- Table 1: difficult to understand what “if negative: none” means.

Thank you, we edited the text using the expression “no further testing”.

- L95 the most studied: really? More studied than NT? Need a reference.

Thank you for the comment. We wrote “one of the most studied”.

Reference: Moreno-Cid M, Rubio-Lorente A, Rodríguez MJ, Bueno-Pacheco G, Tenías JM, Román-Ortiz C, Arias Á. Systematic review and meta-analysis of performance of second-trimester nasal bone assessment in detection of fetuses with Down syndrome. Ultrasound Obstet Gynecol. 2014 Mar;43(3):247-53. doi: 10.1002/uog.13228. PMID: 24151178.

- Table 2: Need references.

We agree with the reviewer’s comments that these figures could be made clearer. We have adjusted figure 2 to accommodate this suggestion.

- L109 Hirshhorn: typo

Thank you, we edited the text with Wolf-Hirshhorn (WHS).

- L116 “echogenic intracardiac focus” seems more frequently used than “intracardiac echogenic focus”. Keep consistency throughout the manuscript.

Thank you for this suggestion. We edited the text accordingly.

- L165 description of NT measurement seems confusing. 

Thank you for the comment. We used standard definition shared by ACOG and ISUOG.

- L195 despite: what?

We thank the reviewer for identifying this typo. This has been corrected.

- L277 posterior risk: a technical term in Bayesian theory would not be reader-friendly; require a more easily expression to readers.

We thank the reviewer for pointing out this error, which has now been corrected. We changed the term “posterior” with “previous”.

Round 2

Reviewer 4 Report

I appreciate response and revision by the authors. Now most of discussion has been settled, and the manuscript deserves publication in Reprod Med.

Please note:

-         EIF still appears as IEF at the heading.

-         There are still some typos, including the name of Hirschhorn, which must be amended at proofreading.

This manuscript is a resubmission of an earlier submission. The following is a list of the peer review reports and author responses from that submission.